
# Evaluating wind profiles in a numerical weather prediction model with Doppler lidar

Pyry Pentikäinen[1], Ewan J. O'Connor[2,3], and Pablo Ortiz-Amezcua[4,5]

[1]Institute for Atmospheric and Earth System Research / Physics, Faculty of Science, University of Helsinki, Helsinki, Finland
[2]Finnish Meteorological Institute, Helsinki, Finland
[3]Department of Meteorology, University of Reading, United Kingdom
[4]Institute of Geophysics, Faculty of Physics, University of Warsaw, Warsaw, Poland
[5]Andalusian Institute for Earth System Research, Granada, Spain

**Correspondence:** Pyry Pentikäinen (pyry.pentikainen@helsinki.fi)

**Abstract.** We use Doppler lidar wind profiles from six locations around the globe to evaluate the wind profile forecasts in the boundary layer generated by the operational global integrated forecast system (IFS) from the European Centre for Medium-range Weather Forecasts (ECMWF). The six locations selected cover a variety of surfaces with different characteristics (rural, marine, mountainous urban, coastal urban).

We first validated the Doppler lidar observations at four locations by comparison with collocated radiosonde profiles to ensure that the Doppler lidar observations were of sufficient quality. The two observation types agree well, with the mean absolute error (MAE) in wind speed almost always less than $1 \ \mathrm{ms^{-1}}$. Large deviations in the wind direction were usually seen only for low wind speeds, and is due to the wind direction uncertainty increasing rapidly as the wind speed tends to zero.

    Time-height composites of the wind evaluation with one-hour resolution were generated and evaluation of the model winds

showed that the IFS model performs best over marine and coastal locations, where the mean absolute wind vector error was usually less than $3 \ \mathrm{ms^{-1}}$ at all heights within the boundary layer. Larger errors were seen in locations where the surface was more complex, especially in the wind direction. For example, in Granada, which is near a high mountain range, the IFS model failed to capture a commonly occurring mountain breeze, which is highly dependent on the sub grid-size terrain features that are not resolved by the model. The uncertainty in the wind forecasts increased with forecast lead time, but no increase in the

bias was seen.

    At one location, we conditionally performed the wind evaluation based on the presence or absence of a low-level-jet diagnosed from the Doppler lidar observations. The model was able to reproduce the presence of the low-level-jet but the wind speed maximum was about $2 \ \mathrm{ms^{-1}}$ lower than observed. This is attributed to the effective vertical resolution of the model being too coarse to create the strong gradients in wind speed observed.

Our results show that Doppler lidar is a suitable instrument for evaluating the boundary layer wind profiles in atmospheric models.





# 1 Introduction

Weather forecasts affect decision making in several weather sensitive sectors (Ebert et al., 2018). Reliable wind forecasts are important for air and maritime transport (Kojo et al., 2011), the wind energy sector (Roulston et al., 2003; Lew et al., 2011),

and for the prediction of other atmospheric phenomena. The development of fog (Gultepe et al., 2007), transport of pollution (Kurita et al., 1985) and the vertical profile of $PM_{10}$ (Sekuła et al., 2021) are all dependent on the winds in the planetary boundary layer. According to the WMO Rolling Review of Requirements Statement of Guidance "wind profiles at all levels outside the main populated areas are a top priority among variables that are not adequately measured by current or planned systems" (Andersson, 2017).

Forecast verification is an important step for understanding both the quality of the forecasts produced, and for the further development of numerical weather prediction (NWP) models. Verification methods are commonly separated into two categories: traditional methods, which consist of various statistics (e.g root-mean-square error, mean absolute error, and standard deviation) calculated by comparing observations to a forecast at specific locations; and spatial methods, which estimate the spatial error in the forecast, (i.e. the forecast has predicted correct features but the location is incorrect (Mass et al., 2002; Casati et al.,

2008)), by estimating how much the forecast must be manipulated to match the observations.

    In recent decades, spatial verification methods have been favoured over traditional verification methods, as the latter cannot take into account spatial errors in the forecast, and thus may cause the forecast to appear much worse than it is in reality. However, spatial verification methods are difficult to use when the quantity being verified is a vector, as is the case for wind, and are typically used only for uni-dimensional variables (e.g. precipitation). While some spatial verification methods have been

adapted for wind forecasts (Skok and Hladnik, 2018), traditional verification methods are still in frequent use (e.g. Pennelly and Reuter, 2017; Olson et al., 2019).

    Wind forecast verification has been performed using various instruments, including radiosondes (Houchi et al., 2010; Fovell and Gallagher, 2020), Doppler radar (Salonen et al., 2008; Beck et al., 2014), in-situ observations (Skok and Hladnik, 2018; Fovell and Gallagher, 2020), and from satellite scatterometer observations (Accadia et al., 2007). Olson et al. (2019) used

Doppler lidar, sodar, radio-acoustic sounding system and radar wind profiler for wind forecast verification to aid NWP development for wind energy use.

    Each instrument has their own benefits and disadvantages. In-situ observations tend to have dense networks giving good horizontal coverage, but the observations are limited to the surface; vertical profiles can be obtained from tall masts, but these are much more sparsely located. Radiosondes are able to capture profiles that encompass the whole troposphere but are usually

launched only twice a day and hence lack temporal resolution. Observations from orbiting satellites also have poor temporal resolution and the measurement uncertainties tend to be high in relation to ground-based instruments (Accadia et al., 2007). Weather radars can provide radial winds with good spatial and temporal resolution (e.g. Holleman, 2005) but require the presence of hydrometeors to make the measurement.

    Doppler lidars provide wind profiles with good temporal and vertical resolution within the planetary boundary layer (Pearson

et al., 2009; Päschke et al., 2015; Pichugina et al., 2017; Nijhuis et al., 2018), which can be used to create diurnal composites of





verification metrics and allows for examination of transitions in diurnal wind patterns. In this paper we evaluate wind forecasts by the Integrated Forecasting System (IFS) of the European Centre for Medium-Range Weather Forecasts (ECMWF) in six locations with varying surface characteristics against winds retrieved by a Doppler lidar. In section 2 we provide details of the measurement stations, the Doppler lidars, and ECMWF IFS. In section 3 we describe the processing applied to the data and

discuss the selected verification metrics, with the evaluation results then presented in section 4.

## 2  Data

### 2.1  Stations

Six stations with long-term Doppler lidar wind observations were selected from around the globe (see Fig. 1) for comparison with the operational IFS model winds. These stations have different surface characteristics, ranging from marine, rural, to

urban and coastal-urban. Four of the stations were operated by the Atmospheric Radiation Measurement program (ARM, Mather et al., 2016) of the U.S. Department of Energy, one by the Finnish Meteorological Institute (FMI), and one by the Andalusian Institute for Earth System Research (IISTA-CEAMA). The six stations selected were:

**SGP:** Southern Great Plains (SGP), US, operated by ARM. Located in central USA, near the border of Oklahoma and Kansas. The area is flat and rural, consisting mainly of cattle pasture and wheat fields. The SGP atmospheric observatory is the

worlds largest climate research facility with an extensive history of atmospheric measurements.

**Darwin:** Australia, operated by ARM. Located on the tropical northern coast of Australia, the station is on the southern edge of Darwin International airport within the city of Darwin (population 130 000, 2011). The local building and canopy height is rather shallow and the landscape is mostly flat.

**Cape Cod:** US, operated by ARM during the Two-Column Aerosol Project. Located in Massachusetts on the northeastern

coast, the cape is shaped like a hook, protruding in the Atlantic ocean first to the east and then turning to the north. The station was close to the north eastern shore of the hook, separated by 40 km from the mainland by Cape Cod Bay, and while the station is connected to the mainland, it is mostly surrounded by water.

**Graciosa:** Azores, Portugal, operated by ARM. The station is located on an airfield on the northern coast of Graciosa Island. The island is part of the Azores island group located in the middle of the Atlantic Ocean, 1600 km west from Portugal.

The island is small, having a diameter around 10 km.

**Granada:** Spain, operated by the Andalusian Institute for Earth System Research (IISTA-CEAMA). Located in the city of Granada (population 232 000, 2018), which is at an altitude of approximately 700 m above sea level (asl), with the Sierra Nevada, the highest mountain range in continental Spain with peaks above 3500 m asl, nearby to the southeast. Various valleys from the mountain range affect the wind flow in the city, making it spatially very heterogeneous. The Doppler

lidar is located on top of a three story building on the western side of the city.





**Table 1.** Measurement stations and observation periods. Nominal data coverage refers to the proportion of time during the measurement periods that both Doppler lidar and IFS data are available before filtering for data quality.

| Station | Coordinates | Surface characteristics | Period | Nominal data coverage |
|---|---|---|---|---|
| SGP | (36.61° N, 97.49° W) | Continental, flat | 28 November 2012 – 20 April 2020 | 88.8% |
| Darwin | (12.43° S, 130.89° E) | Coastal, tropical | 21 September 2012 – 26 June 2014 | 89.1% |
| Graciosa | (39.09° N, 28.03° W) | Marine | 21 October 2014 – 31 December 2019 | 86.0% |
| Cape Cod | (42.03° N, 70.05° W) | Coastal | 16 August 2012 – 19 June 2013 | 99.4% |
| Granada | (37.16° N, 3.61° W) | Urban, mountainous | 3 May 2016 – 11 February 2020 | 54.8% |
| Kumpula | (60.33° N, 25.60° E) | Urban, coastal | 10 April 2018 – 30 September 2020 | 88.3% |

**Kumpula:** Finland, operated by the Finnish Meteorological Institute (FMI). Located on the southern coast of Finland within the Helsinki metropolitan area (population 1.5 million). The station is within 5 km of the Baltic Sea to the south and the local coastline contains a small archipelago.

Measurement periods and nominal data coverage for each station are listed in Table 1.

## 2.2 Doppler lidar

The observed wind profiles were obtained from Halo Photonics Streamline and Streamline XR Doppler lidars, which are commercially available heterodyne pulsed systems capable of full-hemispheric scanning (instrument specifications given in Table 2). These instruments have a nominal maximum range of 9 km or more, and were operated at temporal resolutions of 1-5 s. Scan schedules for each Doppler lidar comprised sequences of conical scans at constant elevation (Velocity Azimuth display, VAD) interspersed with extended periods of vertical stare and other scan types. For this study, we used the vertical profiles of horizontal winds derived from the VAD scans, which were at elevation angles from the horizontal of 60, 70 or 75 degrees, depending on the station.

The standard ARM Doppler lidar scan schedule provides one VAD scan at 60 degrees in elevation from horizontal every 15 minutes, with each VAD scan comprising 8 beams equally-spaced in azimuth and each beam integrating 30 000 pulses. For Granada, the Doppler lidar scan schedule provides one VAD scan at 75 degrees in elevation from horizontal every 10 minutes, with each VAD scan comprising 6 (later 12) beams equally-spaced in azimuth with each beam integrating 30 000 (later 45 000) pulses. For Kumpula, the scan schedule provides one VAD scan at 70 degrees elevation from horizontal every 15 minutes, with each VAD scan comprising 24 beams equally-spaced in azimuth with each beam integrating 45 000 pulses.

For the ARM stations, we used the ARM Doppler lidar wind value-added-product (DLPROFWIND4NEWS, Newsom et al., 2015), and for Granada and Kumpula, the winds were calculated using the method of Päschke et al. (2015).



**Table 2.** Halo Photonics Streamline and Streamline XR heterodyne Doppler lidar specifications.

| | |
|---|---|
| Wavelength | 1.5 $\mu$m |
| Pulse repetition rate | 15 kHz |
| Nyquist velocity | 19.8 m s$^{-1}$ |
| Sampling frequency | 50 MHz |
| Points per range gate | 10 |
| Range resolution | 30 m |
| Pulse duration | 0.2 $\mu$s |
| Divergence | 33 $\mu$rad |
| Antenna | monostatic optic-fibre coupled |

## 2.3 The Integrated Forecast System (IFS)

We use the Doppler lidar-retrieved wind profiles to evaluate wind forecasts generated by the European Centre for Medium-Range Weather Forecasts (ECMWF) from their operational Integrated Forecast System (IFS). The IFS is a global numerical weather prediction (NWP) system assimilating a wide range of observations and generates a range of forecast products, from medium range to seasonal predictions, and both deterministic and ensemble forecasts. In this study we only consider the high resolution deterministic medium-range forecast (referred to as HRES), which currently has a horizontal resolution of approximately 9 km and 137 levels in the vertical. The vertical grid spacing is non-uniform and below 15 km varies from 20 to 300 m with higher resolution closer to the ground. The temporal resolution of the model output is one hour and forecasts up to 10 days in length are run every 12 hours.

The assimilation system ingests wind observations from surface-based instrumentation over land (10 m observations from synoptic stations) and sea (ships and buoys), aircraft, radiosondes, radar wind profilers, weather radars and satellites, with wind speed and direction observations being transformed into wind components (u and v). Satellite observations include ocean scatterometer data, processed using dedicated observation operators and assumed to be equivalent to winds at 10 m above the surface Hersbach (2010). Atmospheric motion vectors (AMV) derived from satellite-observed cloud advection are assimilated as single-level wind observations (default operation, Bormann et al., 2003).

It should be noted that the IFS is under constant development, with a new version typically becoming operational every 6–12 months. We do not aim to investigate how changes to the IFS affect the wind forecasts but there have been some potentially significant model updates during the evaluation period; at the end of June 2013, the number of vertical levels was increased from 91 to 137, and, in March 2016, the horizontal grid was changed from a cubic-reduced Gaussian grid to an octahedral-reduced Gaussian grid, resulting in an increase in horizontal resolution from 16 km to 9 km. A full description of the IFS and





the model updates over time can be found from ECMWF documentation: https://www.ecmwf.int/en/forecasts/documentation-and-support/changes-ecmwf-model/ifs-documentation.

For this study we were provided with the model winds for the nearest model grid point to the station from day-ahead forecasts, which have been initialised at 12:00 UTC the previous day and correspond to forecast hours t+12 to t+35.

## 3 Methodology

### 3.1 Data processing

Except for very close to surface, the IFS has a coarser vertical resolution than the Doppler lidar. Therefore, prior to the evaluation, the IFS winds were interpolated to match the vertical resolution of the Doppler lidar. To ensure that any differences observed between the data sets were due to the performance of the IFS only, the Doppler lidar data was filtered rather strictly to ensure that only data with low uncertainty was included. First, a Signal-to-Noise-Ratio (SNR) threshold of -20 dB was applied. Second, we applied a speckle filter to remove isolated pixels that remained after applying the SNR-filter. The effect of this speckle filter was comparatively small.

Third, all data points having a wind fit residual (Newsom et al., 2015) greater than $1 \, \mathrm{m \, s^{-1}}$ were discarded. The value of the residual is a good indicator of turbulence, and as the wind retrieval assumes horizontal homogeneity (Päschke et al., 2015), wind retrievals with a high residual value cannot be considered reliable.

In some cases, precipitation can adversely affect the Doppler lidar wind retrievals, for example, when the precipitation intensity is not spatially homogeneous. While most of these cases were captured by the filters presented above, there were a few precipitation periods which were not adequately filtered. These were identified using a simple rain filter, which detected whether there were downdrafts exceeding $2 \, \mathrm{m \, s^{-1}}$ in the lowest 200 m of the Doppler lidar vertical velocity profile. If so, the entire profile was discarded. This is obviously not an accurate method for identifying the presence of precipitation, but was sufficient for detecting cases where precipitation was likely to compromise the wind retrieval and had not been removed by the other filters.

The order of the filters only matters for the speckle filter, which must be applied after the SNR-filter.

While the filtering of strong turbulence and precipitation is necessary to achieve consistently high data quality, it also removes the impact of these regimes from the results. Therefore, while we are able to evaluate the model in low-turbulent, non-precipitating cases, it is important to remember that we are not able to evaluate the IFS performance in strongly-turbulent or precipitating situations.

Figure 2 displays an example of the impact of applying these filters in terms of a diurnal composite, showing the average fraction of data that would be removed by each filter at Granada. The residual filter and SNR filter each remove the largest fraction of data, with the patterns mostly resembling each other. This is because the increasing uncertainty in radial winds at low SNR would also result in a large residual after attempting the wind retrieval. There are some differences between these two filters; the SNR filter for this instrument and location becomes stronger more rapidly above 1 km than the residual filter, but the residual filter is more active during the daytime boundary layer at altitudes below 1 km. As the residual filter produces





similar results to the SNR filter, with the additional benefit of identifying retrievals suffering from strong turbulence, some wind
retrieval methods use a residual filter instead of a SNR threshold (Päschke et al., 2015). Here we use both filters in tandem in
order to ensure good data quality. The precipitation filter only removes relatively few profiles, as was intended, but these are
the profiles that are expected to have the largest errors. The speckle filter, which is applied after the SNR filter, mostly removes
some data points above the boundary layer, and generally removes an extra 3-6% of data in this region. Figures 2a-d show the
fraction of data at the nominal scan temporal resolution removed by each filter.

After filtering, the Doppler lidar wind profiles (resolution 10 to 15 minutes) were averaged to match the 1 hour temporal
resolution of the wind profiles from IFS. Averaging was performed separately for the u and v components, from which wind
speed and direction were then calculated. Figure 2e shows the resulting impact of all filters in terms of the 1-hour average
being compared to IFS.

The fraction of filtered data remaining varies from station to station and is dependent on, among others, the dynamics of the
boundary layer, aerosol concentration and the specific Doppler lidar configuration (Frehlich and Kavaya, 1991; Pentikäinen
et al., 2020). The dynamics of the boundary layer determine both the altitude to which sufficient aerosol is lofted, but also the
variation in the strength of turbulence. The SNR is determined by the aerosol concentration and the Doppler lidar configuration.

## 3.2   Validation of Doppler lidar winds

The four ARM stations launch radiosondes operationally, which provided an ideal opportunity to validate the Doppler lidar
wind retrievals. To compare the winds from radiosonde and Doppler lidar, we performed the filtering of the Doppler lidar
as described in section 3.1 and then linearly interpolated the high resolution radiosonde wind profile (about 10 m vertical
resolution) to match the vertical resolution of the Doppler lidar wind profile. We chose four altitudes for the comparison:
117 m above ground level (agl), 247 m agl, 507 m agl, and 1000 m agl.

Figure 3 shows that, for all locations, there are slightly larger differences between the two instruments at the lowest altitude
selected (117 m agl) than above. This was expected as the radiosonde launch may still be impacting the balloon track, and
any difference between Doppler lidar location and radiosonde launch location would have the largest impact near the surface.
The observed winds agree well everywhere except for close to the surface at Darwin. At 117 m agl in Darwin, the radiosonde
reports consistently higher wind speeds than the Doppler lidar. This is thought to be due to the larger distance between the
Doppler lidar and radiosonde launch location, and the change in surface characteristics; the radiosonde was launched next to
the airfield runway, whereas the Doppler lidar was in the vicinity of trees and buildings.

Except for 117 m agl in Darwin, the mean absolute difference in wind speed is $< 1 \, \mathrm{m \, s^{-1}}$. The mean absolute difference
in wind direction was $< 10$ degrees for every site except Darwin. We attribute this to the large proportion of low wind speeds
observed at Darwin; Fig. 3 shows that if low wind speeds $< 3 \, \mathrm{m \, s^{-1}}$ were ignored, the agreement would also be much better
at Darwin. Thus, we are confident in the filtered winds retrieved from Doppler lidar.

Kumpula and Granada lack co-located operational radio soundings so it was not possible to perform the validation of Doppler
lidar winds in the same manner. However, the Doppler lidars at these stations were of similar type and configuration, therefore
we are also confident in the Doppler lidar winds retrieved from these stations.





## 3.3 Comparison metrics

**Table 3.** Evaluation metric definition

| Metric | Symbol | Perfect value | Range | Explanation |
|---|---|---|---|---|
| Wind speed bias | B | 0 | $-\infty, \infty$ | Average wind speed error |
| Mean absolute wind speed error | $\mathrm{MAE_S}$ | 0 | $0, \infty$ | Mean magnitude of wind speed error |
| Mean absolute wind direction error | $\mathrm{MAE_D}$ | 0 | $0, 180$ | Mean magnitude of wind direction error |
| Mean absolute wind vector error | $\mathrm{MAE_V}$ | 0 | $0, \infty$ | Mean magnitude of wind vector error |

For model evaluation, we calculated 4 different metrics; wind speed bias B (observation minus model forecast), mean absolute wind speed error $\mathrm{MAE_S}$, mean absolute wind direction error $\mathrm{MAE_D}$, and mean absolute wind vector error $\mathrm{MAE_V}$. The possible ranges for each of the metrics are listed in Table 3. For the calculation of $\mathrm{MAE_D}$, data with wind speed less than $2 \mathrm{~m~s^{-1}}$ was ignored, as the wind direction uncertainty increases rapidly as the wind speed tends towards $0 \mathrm{~m~s^{-1}}$ (e.g. Newsom et al., 2017).

## 4 Results and discussion

Probability density functions of $\mathrm{MAE_V}$ at each station are presented in Fig. 4. The distribution of $\mathrm{MAE_V}$ includes all winds remaining below 2500 m after filtering as described in section 3.1. The shape of the distributions follow the Weibull distribution and show how the general performance of the IFS changes with respect to the surface type: being best at Graciosa which is marine, followed by the coastal stations of Cape Cod and Kumpula, and becoming poorer at Darwin and especially over the continental surface at SGP and more mountainous region of Granada. However, Fig. 4 does not give any insight into which regimes may be responsible for the poor performance at certain locations.

### 4.1 Seasonal variation

Two stations, SGP and Graciosa, had sufficiently long observation time series to investigate the seasonal variability of the absolute wind vector error. Figure 5 shows a time series of the 30-day moving averages of the absolute wind vector error over SGP at different altitudes. The mean is consistently higher than the median, which is a result of the absolute wind vector errors following a Weibull distribution, but otherwise the mean and median show a similar pattern.

A seasonal cycle for the absolute wind vector error can be seen, particularly at the lower 3 altitudes selected, with larger errors seen during summer and lower errors seen during winter. However, there is significant variation from year to year. There is no clear signal in absolute wind vector error that could be attributed to the upgrade in the resolution of the IFS.





Fig. 6a, shows the monthly $\text{MAE}_\text{V}$ from SGP, which clarifies the seasonal cycle suggested by Fig. 5. Again, the win-

ter/summer contrast is obvious in the 3 lowest altitudes selected. At 1 km agl, the monthly $\text{MAE}_\text{V}$ follows the same pattern from April to September, but displays elevated $\text{MAE}_\text{V}$ in winter, in contrast to the lower altitudes.

The time series of the absolute wind vector error from Graciosa (see Fig. A3) also displays a seasonal cycle, although weaker than at SGP. This cycle can be seen in Fig. 6b, where the monthly $\text{MAE}_\text{V}$ is largest during winter and lower during summer at all altitudes selected.

## 4.2   Composites of absolute wind vector error and wind speed bias

The vertical and temporal resolution of Doppler lidar wind profiles enable the generation of diurnal composites of $\text{MAE}_\text{V}$ and B for each station, as shown in Figs. 7 and 8. These metrics were only calculated for time-height intervals containing at least 50 valid data points.

The degradation of the 12Z t+12-35 wind forecast over time can be seen in $\text{MAE}_\text{V}$ at almost all sites, with the improvement

seen for the new forecast coming at 00Z (approximate local solar time: Graciosa, 23; Kumpula, 2; Cape Cod, 19; SGP, 17; Darwin, 9; Granada, 0). The forecast degradation with time is also visible in $\text{MAE}_\text{S}$ and $\text{MAE}_\text{V}$ (see Figs. A2 and A1), but notably, no degradation in terms of B with respect to the forecast time is seen.

The marine and coastal locations, Graciosa, Kumpula, and Cape Cod, display the lowest values of $\text{MAE}_\text{V}$ at all altitudes, with values generally $< 3 \text{ m s}^{-1}$. For these locations, the impact of the surface on the winds is relatively small, and satellite

scatterometer observations over the ocean or sea basin have been assimilated.

Graciosa shows little diurnal variation in B; for 90% of the composite, $-0.3 > B > 0.2 \text{ m s}^{-1}$. There is more diurnal influence in B for Kumpula, with negative B extending gradually above the surface during the daytime, indicative of the influence of a growing boundary layer. Kumpula also displays a negative bias of about $-1 \text{ m s}^{-1}$ close to the surface, which we attribute to the impact of the urban surface not being captured adequately within the model grid box. Cape Cod also shows some diurnal

variation; at low altitudes (200-400 m agl), there is a variation in B, with positive values during the night. As seen for Kumpula, there are negative values of B during the day extending from the surface upwards, indicating the influence of a growing boundary layer. Kumpula and Cape Cod are coastal locations, so can experience both shallow marine boundary layers and the deeper boundary layers generated over land.

SGP shows a distinct diurnal pattern where $\text{MAE}_\text{V} < 3 \text{ m s}^{-1}$ within the daytime turbulent boundary layer and can exceed

$4 \text{ m s}^{-1}$ during the night. $\text{MAE}_\text{V} > 4 \text{ m s}^{-1}$ is also seen above the boundary layer. The magnitude of B is very small above and within the upper daytime turbulent boundary layer, even when $\text{MAE}_\text{V}$ is large. The lowest 300 m of the daytime boundary layer shows a positive bias reaching $0.7 \text{ m s}^{-1}$. A large positive bias above $1 \text{ m s}^{-1}$ is seen in the nocturnal boundary layer, which has a maximum at altitudes around 300 m agl and has values exceeding $1.5 \text{ m s}^{-1}$. The nocturnal pattern of larger $\text{MAE}_\text{V}$ and B suggests the presence of a low-level jet (LLJ), which is explored in more detail in section 4.4.

Darwin exhibits a weak diurnal pattern in $\text{MAE}_\text{V}$ with lower values during the morning and the largest values at night. However, there is a distinct feature in B that is present throughout the day; a layer of $B < -1 \text{ m s}^{-1}$ in the lowest 150 m agl with another layer of $B > 0.5 \text{ m s}^{-1}$ above this. This bias matches the discrepancy seen between the Doppler lidar and





radiosonde (Fig. 3) and is likely due to the differences in the local surface experienced by the two measurement types and the surface representation in the model.

For Granada, $\mathrm{MAE_V} > 3\ \mathrm{m\,s^{-1}}$ almost everywhere, and can exceed $4\ \mathrm{m\,s^{-1}}$ above 1 km. Low values of $\mathrm{MAE_V}$ are only seen in the morning boundary layer, where the wind speeds themselves are also generally low (median wind speed here, not shown, is about $2\ \mathrm{m\,s^{-1}}$). If IFS also produces low wind speeds, then $\mathrm{MAE_V}$ will also be small even if the wind direction is not correct. Hence, these low wind speeds explain the magnitude of B also being relatively low close to the surface. Above the surface B becomes more and more negative with increasing altitude, reaching $-1.5\ \mathrm{m\,s^{-1}}$ above 1 km agl. During the

afternoon, B is positive close to the surface, $0.5\ \mathrm{m\,s^{-1}}$, but still decreases with altitude, with B becoming negative above 1 km. This station also shows the challenges of wind verification in weak wind conditions, where the wind direction is much more difficult to predict (see Figs. A1 and A2).

### 4.3 Mountain breeze

In Granada, the presence of the Sierra Nevada mountains and generally low wind speeds create conditions suitable for the
frequent development of nocturnal mountain breezes (katabatic winds, Ortiz-Amezcua et al., 2022). Above the Doppler lidar station in Granada, the mountain breeze can be observed as a shallow layer of weak easterly and south-easterly winds near the surface with the wind direction matching that of the valleys. There are several valleys leading from the mountains to the city and the surrounding areas, and the resulting topography (see Fig. 1) is too complex to be resolved by the IFS model. The smoothed IFS orography would not be expected to capture mountain breezes arriving from valley directions that are not
resolved. However, it is important to know how large the local model forecast errors may be, as katabatic flows, although having low wind speeds, can still impact issues such as the formation of fog (Cuxart et al., 2021) and air quality (Li et al., 2018).

To understand the impact of the mountain breeze on errors in the IFS wind profiles, we separated the evaluation into two classes based on the Doppler lidar wind profiles: cases with a mountain breeze, and cases without. To identify mountain breeze
cases in the Doppler lidar winds, the mountain breeze must satisfy 4 conditions: the mountain breeze must be present between 2 hours before sunset and 2 hours after sunrise, although it may persist beyond this time; the wind direction should be between $55°$ and $145°$ close to the surface; the mountain breeze should be at least 120 m deep; the mountain breeze should have a duration of at least 1 hour.

Figure 9 presents the diurnal composites of $\mathrm{MAE_D}$ for when the mountain breeze is present and when it is not present.
At night, $\mathrm{MAE_D}$ ranges from $60°$ to $85°$ near the surface when the mountain breeze is present, and from $45°$ to $70°$ when the mountain breeze is not present. The daytime $\mathrm{MAE_D}$ is around $40°$. While $\mathrm{MAE_D}$ is larger when the mountain breeze is present, the complex topography and the generally low wind speeds makes forecasting the wind direction in Granada difficult under any circumstances.

IFS forecasts the most common nocturnal wind direction near the surface to be from the south, which matches surface
observations at Armilla Airbase, located 4 km south-west of the Doppler lidar station (Ortiz-Amezcua, 2019). However, in the city of Granada, where the Doppler lidar is located, south is the least prevalent wind direction near the surface.



## 4.4 Low level jets

Low level jets occur frequently at SGP (Song et al., 2005). Reliable LLJ prediction is highly relevant for the wind energy sector in terms of energy production (Gadde and Stevens, 2021) and consequently, profits (Roulston et al., 2003). Additionally,
nocturnal precipitation from thunderstorms generated by elevated convection initiated by LLJs is common in the Great Plains region (Gebauer et al., 2018).

To examine the ability of IFS to capture LLJs at SGP, we separated the evaluation into two different classes based on whether an LLJ was present in the Doppler lidar observations. LLJ detection was performed using the method described by Tuononen et al. (2017). Composites of wind speed bias for LLJ cases and non-LLJ cases are presented in Fig. 10. LLJ cases exhibit a bias
of more than $2 \mathrm{~m~s}^{-1}$ up to about 500 m during the night, in contrast to non-LLJ cases, where the bias rarely exceeds $1 \mathrm{~m~s}^{-1}$.

Composites of mean wind speeds for the Doppler lidar and IFS during LLJ cases are shown in Fig. 11. While IFS can clearly produce LLJs at approximately the correct height, the mean wind speed is roughly $2 \mathrm{~m~s}^{-1}$ lower than observed by the Doppler lidar. This is likely due to the effective vertical resolution of IFS smoothing the wind speed profile and reducing both the peak wind speed value and the amount of shear below the jet compared to what is seen in the observations (Houchi et al., 2010).

## 4.5 Timing errors

Almost all of the largest absolute wind vector errors between IFS and Doppler lidar were due to incorrectly timed frontal passages and storms. While the timing of these fronts and storms may have been out by up to 4 hours, IFS was usually able to produce structures that closely resembled the Doppler lidar observations. Therefore, although the absolute wind vector error could be greater than $20 \mathrm{~m~s}^{-1}$, the IFS may still be considered to be performing rather well. We are limited here to
observations from a single Doppler lidar, and we can detect displacement errors only as a temporal error. Given a network of such observations with suitable spatial coverage, it might be possible to evaluate displacement errors in the forecast.

An example of large momentary forecast error due to incorrect timing is shown in Fig. 12. The u and v components of IFS and Doppler lidar winds show similar features, but with IFS displaying an approximately 4-hour delay. Whether this incorrect timing is significant or not is dependent on the intended use of the forecasts (Casati et al., 2008).

## 5   Conclusions

In this study, we used Doppler lidar observations to evaluate the wind profiles produced by a global weather forecast model. We first validated the wind profiles generated from Doppler lidar observations with co-located radiosonde profiles at four locations, to ensure that the Doppler lidar wind profiles were of sufficient quality after filtering. Radiosonde and Doppler lidar winds were in good agreement, with mean absolute wind speed error almost always less than $1 \mathrm{~m~s}^{-1}$. The only exception
was seen at Darwin for the lowest 120 m of the profile, and was attributed to the Doppler lidar location and radiosonde launch site having different surface characteristics. Discrepancies in the wind direction occur mainly at low wind speeds ($< 3 \mathrm{~m~s}^{-1}$),



which arises from the wind direction uncertainty increasing rapidly with decreasing wind speed, and becoming ambiguous as the wind speed tends to $0 \, \mathrm{m \, s^{-1}}$.

We then evaluated ECMWF IFS wind profiles in the boundary layer with Doppler lidar winds observations at six stations
from around the globe, each having different surface characteristics. IFS performed best at stations largely influenced by the ocean or the sea (Graciosa, Cape Cod and Kumpula), with $\mathrm{MAE_V}$ at all altitudes generally $< 3 \, \mathrm{m \, s^{-1}}$.

IFS errors increased at stations with more complex surface types and local topography, especially in terms of wind direction. In Granada, a station located at the foot of a tall mountain range, IFS failed to capture a commonly occurring katabatic mountain breeze, the formation and direction of which was dependent on the sub grid-size terrain features that the model was not able
to resolve. During the mountain breeze, the mean absolute wind direction error between the Doppler lidar observation and the IFS ranged from $60°$ to $85°$. Unsurprisingly, accurate prediction of winds in locations with complex surface properties require better model horizontal resolution than the global IFS.

The evaluation also showed that IFS was able to reproduce LLJs over SGP with a similar structure to observed, but with the mean wind speed in the core of the jet being approximately $2 \, \mathrm{m \, s^{-1}}$ too low. The wind shear in LLJs below the jet maximum
was also too low. This is attributed to the effective vertical resolution of the IFS being too coarse to create the observed vertical gradients in the wind speed.

The uncertainty in the wind forecasts ($\mathrm{MAE_V}$, $\mathrm{MAE_S}$, $\mathrm{MAE_D}$) increased with forecast lead time, but no increase in the bias, B, was seen. Some seasonal dependence was observed in the IFS forecast accuracy, but there was no obvious signal in the time series that could be attributed to be the change in IFS horizontal resolution in March 2016.

Almost all of the large absolute wind vector errors were due to incorrectly timed frontal passages and storms, where the IFS produced realistic changes in winds but with a temporal error of up to 4 hours. Evaluating these displacement errors in the forecast would require a network of wind profile observations with suitable spatial coverage.

It should be noted that the Doppler lidar observations were filtered to ensure their quality and hence our evaluation may not be fully representative of the IFS performance over all regimes. For example, strongly turbulent periods and precipitating
periods have not been evaluated with Doppler lidar observations. For these regimes, observations must be acquired by other instrument types.

*Data availability.*

The Doppler lidar wind profiles and radiosonde profiles for ARM stations are available from the Atmospheric Radiation Measurement (ARM) User Facility portal (ARM, 2013; 2014). Doppler lidar wind profiles from Granada (UGR, 2022) and
Kumpula (FMI, 2022) are stored on Zenodo. The ECMWF IFS profiles over each station are available from the ACTRIS-Cloudnet Data portal (http://cloudnet.fmi.fi)





*Author contributions.* PP developed the software for the post-processing of the Doppler lidar winds the wind profile evaluation. EJO supervised the project and generated the Doppler lidar winds for one of the stations. PP and EJO conceptualised the project and wrote the manuscript. POA generated the Doppler lidar winds for one of the stations and contributed to the discussion on winds.

*Competing interests.* The authors declare that they have no conflict of interest.

*Acknowledgements.* This research was funded by the Vilho, Yrjö and Kalle Väisälä Foundation. We also acknowledge the support of COST (European Cooperation in Science and Technology) Action PROBE (CA18235), and the ACCC (Atmosphere and Climate Competence Center) flagship funding by the Academy of Finland (no. 337552). ARM Data were obtained from the Atmospheric Radiation Measurement (ARM) user facility, a U.S. Department of Energy (DOE) Office of Science user facility managed by the Biological and Environmental

Research Program. We also acknowledge the University of Granada, IISTA-CEAMA and the Finnish Meteorological Institute for providing their Doppler lidar data products.



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

**Figure 1.** Map displaying the locations of the Doppler lidar observation stations (SGP orange, Cape Cod blue, Graciosa black, Granada cyan, Kumpula green, Darwin magenta), with inset zoomed maps highlighting the most important terrain features (no zoomed map is provided for SGP, which is located in flat plains). For Granada, a topographical map has been selected to highlight the proximity of the mountains and the various valleys leading into the city. The Doppler lidar is located near the western edge of the city, with the eastern edge extending approximately 2 km eastwards to where the mountain valleys begin. The topographical map has been adapted from Granada's Geographical Information Systems website, while other inset maps have been obtained from Open Street Map (© OpenStreetMap contributors 2022. Distributed under the Open Data Commons Open Database License (ODbL) v1.0).



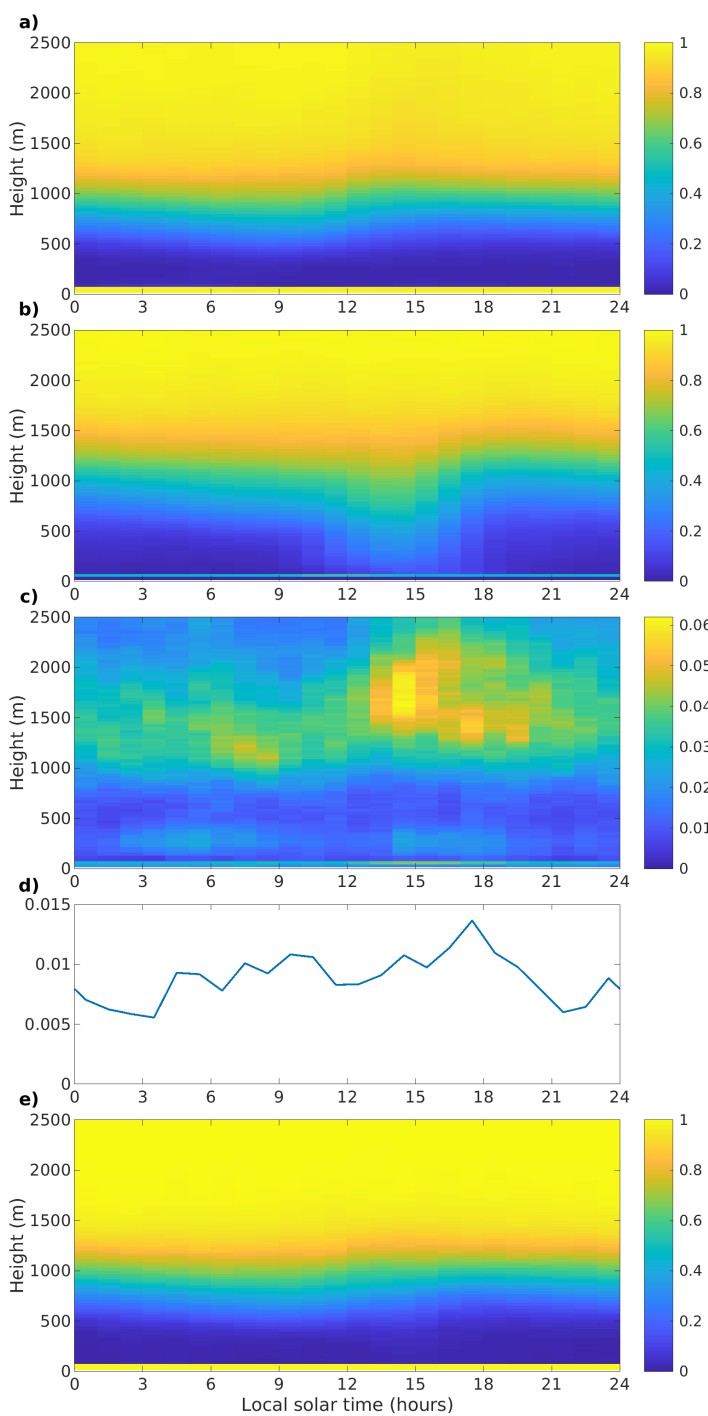

**Figure 2.** Fraction of Doppler lidar winds filtered at Granada by a) SNR filter, b) residual filter, c) speckle filter, d) rain filter, e) all filters combined. Note the change in colour scale for c), since this filter is applied after the SNR filter. The rain filter is applied to the whole column. For panels a)-d) the fraction filtered refers to the winds at nominal temporal resolution (10-15 min), and for e) to the 1-hour averaged data.



**Figure 3.** Scatterplots of Doppler lidar versus radiosonde wind speed and direction at four altitudes and at four locations. The mean absolute difference is also shown in each scatterplot. The wind direction is colour coded in two classes with respect to the wind speed given by the Doppler lidar, denoting whether the wind speed was greater or less than $3\ \mathrm{m\ s^{-1}}$.

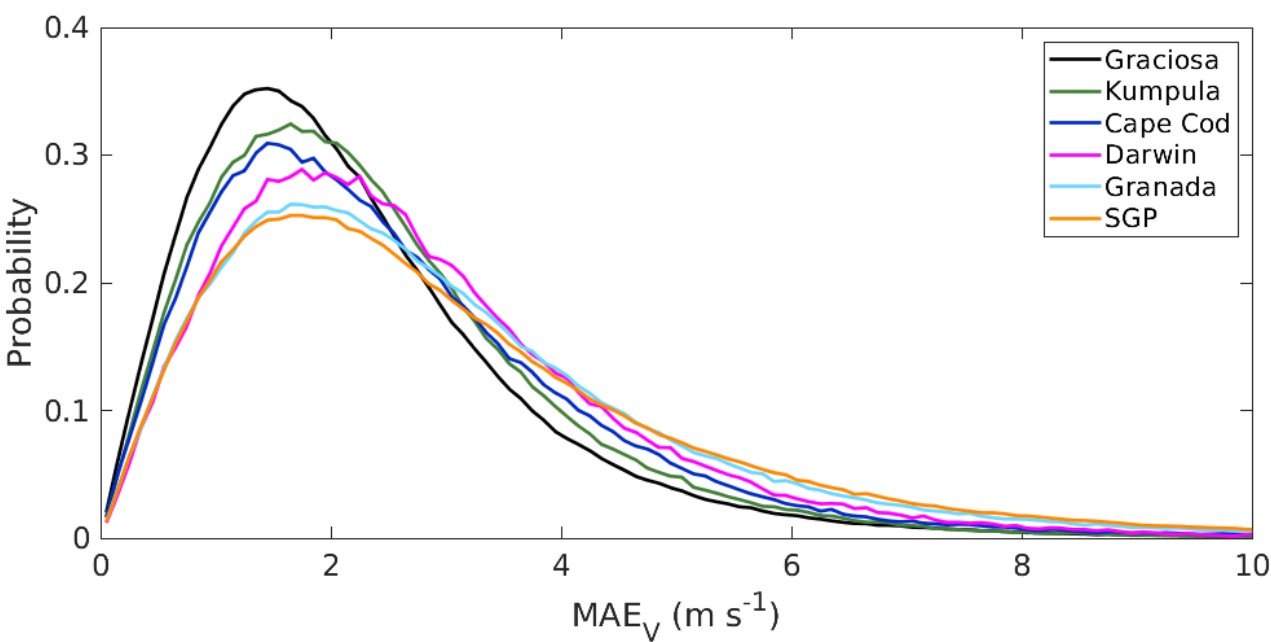

**Figure 4.** Probability density function of the absolute wind vector error at each station.





**Figure 5.** Timeseries of 30-day moving mean (red), median (blue) and 25th and 75th percentiles (black) of wind vector error over SGP at 117 m agl, 247 m agl, 507 m agl, and 1000 m agl.



**Figure 6.** Monthly means of absolute wind vector error at a) SGP and b) Graciosa at 117 m agl, 247 m agl, 507 m agl, and 1000 m agl.





**Figure 7.** Diurnal composites of mean absolute wind vector error at a) Graciosa, b) Kumpula, c) Cape Cod, d) SGP, e) Darwin, and f) Granada.





**Figure 8.** Diurnal composites of wind speed bias at a) Graciosa, b) Kumpula, c) Cape Cod, d) SGP, e) Darwin, and f) Granada.





**Figure 9.** Diurnal composites of mean absolute wind direction error in Granada when a mountain breeze is a) detected, b) not detected, in the Doppler lidar data.





**a)**

**b)**

**Figure 10.** Composites of wind speed bias at SGP for when a) LLJ detected, b) LLJ not detected, in the Doppler lidar data.







**Figure 11.** Composites of a) Doppler lidar and b) IFS mean wind speed at SGP when low level jets were detected in the Doppler lidar data.



**Figure 12.** a) Doppler lidar u wind, b) IFS u wind, c) Doppler lidar v wind, d) IFS v wind, at SGP on 7 August 2013.





**Figure A1.** Diurnal composites of mean absolute wind speed error at a) Graciosa, b) Kumpula, c) Cape Cod, d) SGP, e) Darwin, and f) Granada.







**Figure A2.** Diurnal composites of mean absolute wind direction error at a) Graciosa, b) Kumpula, c) Cape Cod, d) SGP, e) Darwin, and f) Granada.


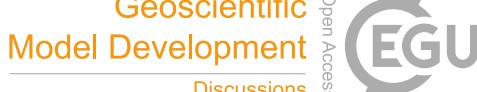

**Figure A3.** Timeseries of 30-day moving mean (red), median (blue) and 25th and 75th percentiles (black) of wind vector error over Graciosa at 117 m agl, 247 m agl, 507 m agl, and 1000 m agl.