# Peer review of "Evaluating wind profiles in a numerical weather prediction model with Doppler lidar"

_Geoscientific Model Development, 2022_

## Referee Comment (RC1)

**Referee's comments to gmd-2022-150**

**General comments**

This paper discusses the results of a comparison of wind profiles between the profiling lidar measurements and numerical weather predictions. The topic is very relevant, and the information provided by remote sensing is extremely valuable to improve the state-of-the-art forecast models.

The literature is fairly reviewed, although there is no mention of the limitations of profiling lidars, i.e. the along-beam average and the assumption of horizontal homogeneity. It is recommended to add a paragraph in the introduction discussing existing studies of errors introduced by complex terrain or front passages on the retrieval of mean quantities from profiling lidars (e.g [1,2]).

The experimental dataset and the model are described thoroughly, and the effort devoted to the development and discussion of the data quality check is commendable. The description of the error metrics would require additional details (see below).

The analysis of the results is interesting, but the section regarding the daily and seasonal statistics of the error can be improved as follows:

- By adding the error on the mean or stricter statistical error check to ensure that the random noise is identified or removed before discussing physical trends
- By expanding the discussion on possible causes of the seasonal variation of the error observed at SGP and Graciosa

Provided that the previous points are addressed (which are deemed as minor reviews for such a comprehensive study), the paper is suitable for publication in this journal, in the Reviewer's opinion.

**Specific comments**

Line 50: Please replace "lack temporal resolution" with "are affected by limited data availability". In fact, the radiosondes can have a good temporal resolution within one single launch.

Lines 132-133, "Therefore…Doppler lidar": please discuss possible errors introduced by the vertical upsampling of simulation data.

Line 135: Please justify the choice of the SNR threshold.

Line 136: Please provide a reference for the speckle filter.

Lines 180-181, "This…surface": It is unclear why the uncertainty in the balloons' position may lead to higher error near the ground, as the sondes generally drift further away at higher altitude due to the stronger winds. If the balloons' launch sites are not collocated with the lidar locations, then the error can be due to the larger horizontal variability of wind speed close to the surface due to the effect of the local terrain. Please clarify.

Line 195: Please provide an explicit formula for the mean absolute wind vector error.

Figure 6: Please add error bar equal to the monthly standard deviation to give a sense of the inter-annual variability.

Figures 7, 8: In some cases (Kumpula, Cape Cod) the daily patterns do not emerge significantly on top of the statistical noise. Please add a colormap with the error on the mean of the error metrics (see [3]), or reject data affected by an error on the mean higher than a reasonable threshold.

Figure 8, 10: Please use a colormap suitable for data containing positive and negative values (e.g. BlueWhiteRed in Matlab, ReBu in python) for easier readability.

Lines 237-238, "Kumpula…land": this statement could be substantiated by plotting the dependence of the error on the wind direction.

Line 288: Please correct the typo "an LLJ".

**References:**

[1] Bingöl, F. (2009). Complex terrain and wind lidars.

[2] Klaas-Witt, T., & Emeis, S. (2022). The five main influencing factors for lidar errors in complex terrain. Wind Energy Science, 7(1), 413-431.

[3] Zięba, A., Ramza, P. (2011). Standard deviation of the mean of autocorrelated observations estimated with the use of the autocorrelation function estimated from the data. Metrology and Measurement Systems, 18(4), 529-542.

---

## Referee Comment (RC2)

**Summary**

The authors present an evaluation study of wind profile forecasts in the boundary layer, from the Integrated Forecasting System (IFS) of the European Centre for Medium-Range Weather Forecasts (ECMWF), against winds retrieved by Doppler Lidars. Lidar data from 6 locations with different characteristics are used, while the retrievals are primarily validated using collocated radiosonde profiles. Traditional methods, consisting of various statistics, such as the calculation of mean absolute errors and wind speed bias, are used to compare observations with forecast. The authors show that IFS winds are more accurately predicted in marine and coastal locations (Graciosa, Cape Cod and Kumpula), while errors increase at stations with more complex topography. Moreover, the authors present statistics on seasonality and discussion on the effect of local characteristics (Low Level Jet at SGP, Katabatic winds at Granada).

The study is very relevant to the journal and the results are valuable to improve the state-of-the-art forecast models. The manuscript is well-written, carefully organized and provides the right amount of detail for the reader. The authors conducted a study with novelty and clarity; thus, I recommend for its publication, with only minor comments and revision (see below).

**General comments**

The manuscript mainly discusses the difference between the wind lidar measurements and the modeled ones. In the relevant plots, only the biases are plotted and discussed, without a mention on the mean wind values, which could be used as a reference of the importance of the absolute bias. Can the authors explain why this is not included? The authors could consider to include in the manuscript one of the following: (a) the mean wind speed values (by the model or the lidar), when they discuss their absolute errors/biases; (b) a scatterplot of the lidar vs the modeled wind speeds for the different stations and altitudes (similar as Figure 3); or (c) a diurnal colorplot of the mean wind speed for the different stations (similar as Figure 7).

A math appendix with the equations used for the calculation of the different parameters and their errors would be useful to the readers (e.g. for the wind speed and direction calculated from u and v components, the B, $MAE_S$, $MAE_D$ and $MAE_V$).

It is good to mention somewhere in the manuscript that (a) all elevations in the plots are above surface level and (b) the authors used the IFS outputs, which are above surface elevation, to compare with the relevant lidar measurements. If this is the case.

**Specific comments**

At Lines 68-88, and in table 1, the 6 stations are described. It would be of interest to the readers to include the elevation above sea level of the Lidars (e.g. in table 1, next to the coordinates). The terrain altitude variability around the stations, in the IFS resolution, would be an interesting information also.

Lines 96-104: Could you comment on the reason why in different locations the VAD scans were scheduled in different angels? Is it related with the vertical resolution and max altitude of interest in each site, or something else?

Lines 109-110: The syntax can be improved to read the sentence better. Consider revising to this end.

Lines 110-113: Please include a reference on the ECMWF IFS HRES model used. Furthermore, some additional information for the model could be interesting for the reader. Namely: (a) how many (approx.) of the 137 vertical levels are below 1km, since PBL winds are discussed in this study, (b) what is the range of the vertical grid spacing in the PBL heights, (c) the information that IFS HRES runs on a Gaussian grid or (if available) the horizontal resolution in degrees.

Lines 115-117: Consider including a reference on these assimilations.

Line 136: Speckle filter: add a reference on this filter used.

Lines 143- 145: Rain filter: in case this filtered is used in the past for another study (known to the authors), they could include a reference of prior use when describing this filter.

Figure 2: I suggest to present the different sub-plots with the same order as discussed in the manuscript (so the speckle filter (c ) to be before the residual filter (b)). Additionally, consider including titles at the colorbars (and at y-axis in d) or subtitles with the information of each filter in the subplots.

L 175-177: It would be good if the resolution of Wind Lidar is also mentioned here, to highlight the difference between Radiosonde (10m) and Lidar (30m) resolution.

Figure 3: It would be of interest to add the station names above the 4 columns of the doppler lidar wind direction plot.

Section 3.2: It would be interesting to include the information of the distance between the Radiosonde Launch points and the Wind Lidars for the 4 sites used.

Line 180: "This was expected as the radiosonde launch may still be impacting the balloon track". This sentence is not clear. Please rephrase to include why this is more relevant in the lower altitudes.

Lines 186-189: At 117m, in Darwin, all wind speeds bellow 15m/s (not only the ones bellow 3m/s) have a significant different between the lidar and the radiosonde (in comparison to the differences in the other stations/altitudes). So please include a quantification of the agreement discussed in the sentence: "Fig. 3 shows that if low wind speeds < 3 m s−1 were ignored, the agreement would also be much better at Darwin".

Line 211–212: Could you comment on the reason why the errors are larger during summer and lower during winter for SGP? At Line 217, you mention a weaker seasonal cycle for Graciosa. It is also an opposite cycle, with larger errors during winter and lower during summer. Could you comment on the reason for this contradiction? Is this connected with relatively higher/lower wind speed values in the periods of higher/lower MAEv in the two sites? Did you investigated also the Mean Absolute Percentage Error (MAPE) for these two sites, and if so, do you observe the same seasonal variability?

Figure 6: Please consider adding error bars in this plot.

Figures 7 and 8: It would be helpful for the reader if you can include sub-legends with each station name/location in the sub-plots. Also, a symbol indicating the local solar time in each site discussed in the manuscript) would be useful.

Line 236: "...indicating the influence of a growing boundary Layer". Do you imply that the growing boundary layer is not correctly captured from the model, or that the PBL winds are modeled with higher biases? Please specify if possible.

Lines 265-266: Please improve this sentence to read better. Maybe break it to two smaller ones.

L 288-289: Can the authors briefly describe the method according to which they detected LLJ? The reference provided, shows the clarity of the data processing, but it would be easy for the reader to have a short briefing of the method.

**Technical comments**

Line 119: "surface Hersbach (2010)": correct the reference syntax.

Line 158: "filter is more active": consider revising to "more effective".

In Figure 8: A diverging colorbar separating the negative from the positive values of the wind speed bias (for example red=negative, white=0, blue=positive) would be really helpful for the reader. This way, it would be easier to point out the diurnal evolution characteristics of PBL mentioned at 231-238.

Lines 113-114: Please improve the syntax of this sentence.

Curtain plots: What does the white space at the bottom of each curtain plot represents? Is it the height that Wind Lidar starts to measure? If so, it would be interesting to include this information in the general description of the systems.

Figure 12: Please add labels: "Wind lidar" for (a), (c) and "IFS" for (b), (d). Here, a diverging colorbar would also be very helpful.

Line 323: "... to observed": I think "to" need to be omitted.

Figures A1, A2: Please add labels with the locations.

---

## Author Comment (AC1)

**Response to referees comments**

September 2022

We have addressed all of the points raised by the reviewers (copied here and shown in black text), and include our responses to each point below (in blue text). Where applicable, we also provide the new text (in blue italics).

**1 RC2 Eleni Marinou**

The manuscript mainly discusses the difference between the wind lidar measurements and the modeled ones. In the relevant plots, only the biases are plotted and discussed, without a mention on the mean wind values, which could be used as a reference of the importance of the absolute bias. Can the authors explain why this is not included? The authors could consider to include in the manuscript one of the following: (a) the mean wind speed values (by the model or the lidar), when they discuss their absolute errors/biases; (b) a scatterplot of the lidar vs the modeled wind speeds for the different stations and altitudes (similar as Figure 3); or (c) a diurnal colorplot of the mean wind speed for the different stations (similar as Figure 7).

We have added a new figure showing diurnal composites of the mean Doppler lidar wind speeds in the appendix.

A math appendix with the equations used for the calculation of the different parameters and their errors would be useful to the readers (e.g. for the wind speed and direction calculated from u and v components, the B, MAES, MAED and MAEV).

An appendix describing the equations for the verification metrics has been added to the manuscript. The following equations have been added. Wind speed bias is given by

$$\mathbf{B} = \frac{1}{N} \sum_{i=1}^{N} (U_{O_i} - U_{F_i}), \tag{1}$$

where N is the number of data points,  $U_O$  is the observed wind speed and

 $U_F$  is the forecasted wind speed. Mean absolute wind speed error is given by

$$MAE_{S} = \frac{1}{N} \sum_{i=1}^{N} |(U_{O_{i}} - U_{F_{i}})|.$$
(2)

Mean absolute wind direction error is given by

$$MAE_{D} = \frac{1}{N} \sum_{i=1}^{N} min(|\delta_{O_{i}} - \delta_{F_{i}}|, |360^{\circ} + \delta_{O_{i}} - \delta_{F_{i}}|, |360^{\circ} + \delta_{F_{i}} - \delta_{O_{i}}|), \quad (3)$$

where  $\delta_O$  is the observed wind direction in degrees and  $\delta_F$  is the forecasted wind speed in degrees. Mean absolute wind vector error is given by

$$MAE_{V} = \frac{1}{N} \sum_{i=1}^{N} |(\mathbf{U}_{\mathbf{O}_{i}} - \mathbf{U}_{\mathbf{F}_{i}})|, \qquad (4)$$

where  $U_O$  is the observed wind vector and  $U_F$  is the forecasted wind vector.

It is good to mention somewhere in the manuscript that (a) all elevations in the plots are above surface level and (b) the authors used the IFS outputs, which are above surface elevation, to compare with the relevant lidar measurements. If this is the case.

This information has been added to the manuscript: All metrics are calculated with respect to the ground level at the Doppler lidar location. Note that the observation and model surface altitude above mean sea level may not agree in locations with locally varying topography.

**1.1** Specific comments**

At Lines 68-88, and in table 1, the 6 stations are described. It would be of interest to the readers to include the elevation above sea level of the Lidars (e.g. in table 1, next to the coordinates). The terrain altitude variability around the stations, in the IFS resolution, would be an interesting information also.

The altitude of the Doppler lidars has been included in Table 1, together with the model surface altitude.

Lines 96-104: Could you comment on the reason why in different locations the VAD scans were scheduled in different angels? Is it related with the vertical resolution and max altitude of interest in each site, or something else?

Except for one instrument, we do not control these Doppler lidars. The Doppler lidar scan schedules are created by the organisations responsible for them (e.g. ARM) and they may have multiple reasons for choosing the VAD elevation angle for a particular site, including the reasons suggested by the reviewer. The choice of VAD elevation angle may have some impact on data availability, but should not have a major impact on the wind retrieval itself, which was the object of this study.

Lines 109-110: The syntax can be improved to read the sentence better. Consider revising to this end.

We have revised this sentence to: The IFS is a global numerical weather prediction (NWP) system, which assimilates a wide range of observations and generates a range of forecast products. The forecast products include medium range to seasonal predictions, and both deterministic and ensemble forecasts.

Lines 110-113: Please include a reference on the ECMWF IFS HRES model used. Furthermore, some additional information for the model could be interesting for the reader. Namely: (a) how many (approx.) of the 137 vertical levels are below 1km, since PBL winds are discussed in this study, (b) what is the range of the vertical grid spacing in the PBL heights, (c) the information that IFS HRES runs on a Gaussian grid or (if available) the horizontal resolution in degrees.

There is no standard journal reference for the ECMWF IFS HRES model. We have provided the link to ECMWF documentation towards the end of this section: https://www.ecmwf.int/en/publications/ifs-documentation which details the continuous updates to this operational forecast model. Additional information on the model has been included: *There are about 20 model levels in* the lowest 1 km, with the vertical resolution ranging from 20 m to 100 m. The text does state that the model uses an octahedral-reduced Gaussian grid, and the resolution (in km) is also provided.

Lines 115-117: Consider including a reference on these assimilations.

There are a large number of observations assimilated, which makes it difficult to select an appropriate reference. All assimilation information is given in the link to the IFS documentation towards the end of this section.

Line 136: Speckle filter: add a reference on this filter used.

The name of the filter was incorrect, and is now called column-wise threshold filter. A description of the filter is also included. First, a Signal-to-Noise-Ratio (SNR) threshold of -20 dB was applied. Second, we applied a column-wise threshold filter, which identified the first point with SNR below the -20 dB threshold, and then removed all data above. Note that these two steps could be combined into one filter, but separating the steps gave the opportunity to identify where there was no longer continuity in the wind profile. The effect of the column-wise threshold filter was comparatively small when applied after the SNR filter.

Lines 143- 145: Rain filter: in case this filtered is used in the past for another study (known to the authors), they could include a reference of prior use when describing this filter.

This filter was developed for this study.

Figure 2: I suggest to present the different sub-plots with the same order as discussed in the manuscript (so the speckle filter (c) to be before the residual filter (b)). Additionally, consider including titles at the colorbars (and at y-axis in d) or subtitles with the information of each filter in the subplots.

The order of the filters has been changed. We decided against including a colourbar title since they are all 'fraction'.

L 175-177: It would be good if the resolution of Wind Lidar is also mentioned here, to highlight the difference between Radiosonde (10m) and Lidar (30m) resolution.

Vertical resolution of the Doppler lidar wind profile is now included in this sentence. Note that the vertical resolution is 26 m at ARM locations (due to 60 degree elevation angle from horizontal).

Figure 3: It would be of interest to add the station names above the 4 columns of the doppler lidar wind direction plot

The station names have been now added to the figure.

Section 3.2: It would be interesting to include the information of the distance between the Radiosonde Launch points and the Wind Lidars for the 4 sites used.

This information has been included in the text.

Line 180: "This was expected as the radiosonde launch may still be impacting the balloon track". This sentence is not clear. Please rephrase to include why this is more relevant in the lower altitudes.

In the first few tens to hundreds of metres the balloon is still accelerating vertically and swinging after launch. Further research into the specific issue at Darwin highlighted that radar was used to track the balloon and provide winds, rather than GPS. We have updated the text as follows: This is unlikely to be due to distance only, as SGP still compares well even though the distance between the Doppler lidar and radiosonde launch location is further. At Darwin, radiosonde winds were obtained by radar tracking rather than GPS during the period involved in this study and it is thought that a mismatch between the stated and actual launch location is partly responsible for the discrepancy close to the surface.

Lines 186-189: At 117m, in Darwin, all wind speeds bellow 15m/s (not only the ones bellow 3m/s) have a significant different between the lidar and the

radiosonde (in comparison to the differences in the other stations/altitudes). So please include a quantification of the agreement discussed in the sentence: "Fig. 3 shows that if low wind speeds < 3ms - 1 were ignored, the agreement would also be much better at Darwin".

This sentence refers to the wind direction difference, where the impact of wind speed to the agreement is significant. For the wind speed, there is a significant difference at all wind speeds. When ignoring the wind speeds below  $3 \text{ ms}^{-1}$ , the mean absolute wind direction differences are  $7.3^{\circ}$ ,  $8.2^{\circ}$ ,  $5.8^{\circ}$  and  $7.7^{\circ}$  respectively for 117 m, 247 m, 507 m and 1000 m.

Line 211–212: Could you comment on the reason why the errors are larger during summer and lower during winter for SGP? At Line 217, you mention a weaker seasonal cycle for Graciosa. It is also an opposite cycle, with larger errors during winter and lower during summer. Could you comment on the reason for this contradiction? Is this connected with relatively higher/lower wind speed values in the periods of higher/lower MAEv in the two sites? Did you investigated also the Mean Absolute Percentage Error (MAPE) for these two sites, and if so, do you observe the same seasonal variability?

We think that larger errors in summer at SGP are due to the presence of more strong convective events in summer; these are harder for forecast models to predict than large-scale synoptic events. Graciosa, being in the middle of the Atlantic, would experience mostly large-scale synoptic patterns whether in summer or winter, hence the weaker seasonal cycle; and the poorer performance winter may be due to stronger winter cyclones. However, such statements are difficult to test from our datasets alone.

Figure 6: Please consider adding error bars in this plot.

We have added bars showing the maximum and minimum monthly average across all years. We do not consider standard deviation based on the hourly data a descriptive metric of the inter-annual variation as the momentary model error can vary significantly depending on the weather conditions.

Figures 7 and 8: It would be helpful for the reader if you can include sublegends with each station name/location in the sub-plots. Also, a symbol indicating the local solar time in each site discussed in the manuscript) would be useful.

The figures have been updated to include legends. The plots are in approximate local solar time; this is now stated in both the captions and in the text. Approximate local solar time with respect to UTC has also been added to Table 1.

Line 236: "...indicating the influence of a growing boundary Layer". Do you imply that the growing boundary layer is not correctly captured from the model, or that the PBL winds are modeled with higher biases? Please specify

**if possible.**

It could be for a number of reasons, including: rate of growth of BL not captured, timing of growth of BL not captured, incorrect model surface roughness producing incorrect wind profile. Observations in a location with a heterogeneous surface may also not be representative of the surrounding area, particularly close to the surface. Hence, we would prefer not to speculate without further research.

Lines 265-266: Please improve this sentence to read better. Maybe break it to two smaller ones.

The sentence is replaced by: However, it is important to know how large the local model forecast errors may be. Even at low wind speeds, katabatic flows can still impact issues such as the formation of fog (Cuxart et al., 2021) and air quality (Li et al., 2018).

L 288-289: Can the authors briefly describe the method according to which they detected LLJ? The reference provided, shows the clarity of the data processing, but it would be easy for the reader to have a short briefing of the method.

A brief description of the method has been included: , which identifies continuous local wind speed maxima in the vertical wind speed profile being both at least 2 m s-1 stronger and at least 25% stronger than local wind speed minima above and below

**1.2** Technical comments**

Line 119: "surface Hersbach (2010)": correct the reference syntax.

Changed to "surface (Hersbach, 2010)"

Line 158: "filter is more active": consider revising to "more effective".

The phrase was changed as suggested.

In Figure 8: A diverging colorbar separating the negative from the positive values of the wind speed bias (for example red=negative, white=0, blue=positive) would be really helpful for the reader. This way, it would be easier to point out the diurnal evolution characteristics of PBL mentioned at 231-238.

A Blue-White-Red colourmap has been applied to figures 8 and 10.

Lines 113-114: Please improve the syntax of this sentence.

This sentence has been revised to: Forecasts up to 10 days in length are initialised every 12 hours, and the temporal resolution of the model output is

**one hour.**

Curtain plots: What does the white space at the bottom of each curtain plot represents? Is it the height that Wind Lidar starts to measure? If so, it would be interesting to include this information in the general description of the systems.

The white space is the blind zone for the Doppler lidar systems. The following sentence has been added to the Doppler lidar description section: Data from the first 60-90 m in range suffers with contamination from the outgoing pulse and is discarded.

Figure 12: Please add labels: "Wind lidar" for (a), (c) and "IFS" for (b), (d). Here, a diverging colorbar would also be very helpful.

The labels have been added, and a diverging colorbar applied.

Line 323: "... to observed": I think "to" need to be omitted.

The word "to" has been omitted.

Figures A1, A2: Please add labels with the locations.

Labels with locations have been added.

**2 RC1 Stefano Letizia**

**2.1 General comments**

The literature is fairly reviewed, although there is no mention of the limitations of profiling lidars, i.e. the along-beam average and the assumption of horizontal homogeneity. It is recommended to add a paragraph in the introduction discussing existing studies of errors introduced by complex terrain or front passages on the retrieval of mean quantities from profiling lidars (e.g [1,2]). The experimental dataset and the model are described thoroughly, and the effort devoted to the development and discussion of the data quality check is commendable. The description of the error metrics would require additional details (see below). The analysis of the results is interesting, but the section regarding the daily and seasonal statistics of the error can be improved as follows:

The following text has been included in the introduction: Most wind retrievals from an individual scanning Doppler lidar assume that the flow is horizontal and that flow is homogeneous, assumptions that may no longer be valid in complex terrain (Bingöl et al., 2009; Klass-Witt and Emeis, 2022), and in strongly turbulent situations (Robey and Lundquist, 2022; Rahlves et al., 2022), hence additional checks must be performed to ensure the quality of the wind retrievals • By adding the error on the mean or stricter statistical error check to ensure that the random noise is identified or removed before discussing physical trends

We have added the range of variation to the seasonal plot. For the diurnal composites, we are confident that after the performed filtering and averaging, the statistical noise is not an issue for the purpose we are studying here.

• By expanding the discussion on possible causes of the seasonal variation of the error observed at SGP and Graciosa

We think that larger errors in summer at SGP are due to the presence of more strong convective events in summer; these are harder for forecast models to predict than large-scale synoptic events. Graciosa, being in the middle of the Atlantic, would experience mostly large-scale synoptic patterns whether in summer or winter, hence the weaker seasonal cycle; and the poorer performance winter may be due to stronger winter cyclones. However, such statements are difficult to test from our datasets alone.

**2.2 Specific comments**

Line 50: Please replace "lack temporal resolution" with "are affected by limited data availability". In fact, the radiosondes can have a good temporal resolution within one single launch.

The sentence has been changed as suggested.

Lines 132-133, "Therefore...Doppler lidar": please discuss possible errors introduced by the vertical upsampling of simulation data

The Doppler lidar and model vertical resolution are quite similar. In addition, the model wind profile is smooth and linear interpolation would only cause issues in strong vertical gradients (i.e. very close to the surface) which is not where we are measuring.

Line 135: Please justify the choice of the SNR threshold.

The velocity uncertainty is directly related to SNR. A reference for the choice of SNR threshold has been added: (Manninen et al., 2018).

Line 136: Please provide a reference for the speckle filter.

The name of the filter was incorrect, and is now called column-wise threshold filter. A description of the filter is also included. First, a Signal-to-Noise-Ratio (SNR) threshold of -20 dB was applied. Second, we applied a columnwise threshold filter, which identified the first point with SNR below the -20 dB threshold, and then removed all data above. Note that these two steps could be combined into one filter, but separating the steps gave the opportunity to identify where there was no longer continuity in the wind profile. The effect of

**the column-wise threshold filter was comparatively small when applied after the SNR filter.**

Lines 180-181, "This...surface": It is unclear why the uncertainty in the balloons' position may lead to higher error near the ground, as the sondes generally drift further away at higher altitude due to the stronger winds. If the balloons' launch sites are not collocated with the lidar locations, then the error can be due to the larger horizontal variability of wind speed close to the surface due to the effect of the local terrain. Please clarify.

The Doppler lidar and radiosonde launch locations are not perfectly colocated, although they are reasonably close at all sites, and within 40 m at 2 sites. As stated by the reviewer, close to the surface, deviations between the two measurements may partially be due to the larger horizontal variability of wind speed close to the surface due to the effect of the local terrain. Additionally, in the first few tens to hundreds of metres the balloon is still accelerating vertically and swinging after launch. Further research into the specific issue at Darwin highlighted that radar was used to track the balloon and provide winds, rather than GPS. We have updated the text as follows: This is unlikely to be due to distance only, as SGP still compares well even though the distance between the Doppler lidar and radiosonde launch location is further. At Darwin, radiosonde winds were obtained by radar tracking rather than GPS during the period involved in this study and it is thought that a mismatch between the stated and actual launch location is partly responsible for the discrepancy close to the surface.

Line 195: Please provide an explicit formula for the mean absolute wind vector error.

Math appendix added with the following equations: Wind speed bias is given by

$$\mathbf{B} = \frac{1}{N} \sum_{i=1}^{N} (U_{O_i} - U_{F_i}), \tag{5}$$

where N is the number of data points,  $U_O$  is the observed wind speed and  $U_F$  is the forecasted wind speed. Mean absolute wind speed error is given by

$$MAE_{S} = \frac{1}{N} \sum_{i=1}^{N} |(U_{O_{i}} - U_{F_{i}})|.$$
(6)

Mean absolute wind direction error is given by

$$MAE_{D} = \frac{1}{N} \sum_{i=1}^{N} min(|\delta_{O_{i}} - \delta_{F_{i}}|, |360^{\circ} + \delta_{O_{i}} - \delta_{F_{i}}|, |360^{\circ} + \delta_{F_{i}} - \delta_{O_{i}}|), \quad (7)$$

where  $\delta_O$  is the observed wind direction in degrees and  $\delta_F$  is the forecasted wind speed in degrees. Mean absolute wind vector error is given by

$$MAE_{V} = \frac{1}{N} \sum_{i=1}^{N} |(\mathbf{U}_{\mathbf{O}_{i}} - \mathbf{U}_{\mathbf{F}_{i}})|, \qquad (8)$$

where  $U_O$  is the observed wind vector and  $U_F$  is the forecasted wind vector.

Text included after line 195: Equations for the metrics are listed in Appendix A.

Figure 6: Please add error bar equal to the monthly standard deviation to give a sense of the inter-annual variability.

Bars showing the maximum and minimum monthly average across all years have been added. We do not consider standard deviation based on the hourly data a descriptive metric of the inter-annual variation as the momentary model error can vary significantly based on the weather conditions.

Figures 7, 8: In some cases (Kumpula, Cape Cod) the daily patterns do not emerge significantly on top of the statistical noise. Please add a colormap with the error on the mean of the error metrics (see [3]), or reject data affected by an error on the mean higher than a reasonable threshold.

For Figure 7, we do not consider there being a discernible diurnal pattern in Kumpula and Cape Cod beyond the degradation of the forecast with time. The forecast data does not have noise related uncertainty, and thus all of the uncertainty in the error metric arises from the Doppler lidar data. The Doppler lidar uncertainty is independent of the forecast error. Therefore uncertainty in the error metrics is the same as the Doppler lidar uncertainty, which based on the filtering described in Section 3.1 should be below 0.5 m/s for data included in the calculation of the mean, and thus the error of the mean should be even smaller.

Figure 8, 10: Please use a colormap suitable for data containing positive and negative values (e.g. BlueWhiteRed in Matlab, ReBu in python) for easier readability.

The colourmap in Figures 8 and 10 has been changed to BlueWhiteRed.

Lines 237-238, "Kumpula...land": this statement could be substantiated by plotting the dependence of the error on the wind direction.

We attempted to generate plots as suggested by the reviewer but felt that these did not add significant value to the article. In the situations described here, the wind direction can change from on-shore to off-shore (i.e. 180 degrees) over the depth of the boundary layer.

Line 288: Please correct the typo "an LLJ".

Typical usage for an initialism is to use the indefinite article "a" or "an" depending on how the term is pronounced, rather than whether the first letter is a vowel or consonant, hence our choice of "an". We suggest leaving this to the copy editor to refine.